# Spatial Layout Planning of Urban Emergency Shelter Based on Sustainable Disaster Reduction

**DOI:** 10.3390/ijerph20032127

**Published:** 2023-01-24

**Authors:** Wenlong Zhu, Houlong Xing, Wenlu Kang

**Affiliations:** 1Department of Urban and rural planning, School of Architecture and Design, China University of Mining and technology, Xuzhou 221116, China; 2Department of Architecture, School of Architecture and Design, China University of Mining and technology, Xuzhou 221116, China

**Keywords:** emergency shelter, central urban area of Xuzhou, ArcGIS, layout optimization

## Abstract

An urban emergency shelter provides a place of temporary life and shelter for victims after a disaster. As a very important public service facility, its spatial layout is greatly related to the security of lives and the property of the urban residents. Upholding the concept of sustainable disaster reduction, this study took the central urban area of Xuzhou as an example. Based on the analysis of ArcGIS software, this study analyzed and planned the spatial layout of emergency shelters in Xuzhou and visualized the service area ratio, service population ratio, service capacity ratio, and service overlap rate of each administrative district. Finally, 73 fixed emergency shelters were determined, among which eight were classified as central shelters. At the same time, through consulting the relevant data, it was found that similar problems such as potential safety hazard, blind areas, service overlapping, and mismatch of shelter layout and actual needs also exist in other cities. Finally, in light of the existing problems, relevant suggestions are provided for the adjustment and optimization of the layout of emergency shelters.

## 1. Introduction

Emergency shelters are important facilities for displacing victims in response to earthquakes, fires, floods, epidemics, and other public emergencies, and are the guarantee of urban emergency response and disaster prevention [1]. To maximize their shelter and relief functions after a disaster, the scientific planning, layout, and maintenance of emergency shelters are some of the key issues that need to be urgently studied in the field of disaster prevention and mitigation in China [2].

Due to accelerating urbanization, and the highly-dense urban population and buildings, all kinds of natural and man-made disasters have become more severe. More and more experts and scholars recognize the great significance of reasonable planning and the construction of emergency shelters for the safety and sustainable development of cities. Yu Lixin proposed building additional shelters and other measures based on the analysis of the layout and traffic accessibility of existing emergency shelters [3]; Chen Mingjie and Lv Fei evaluated the network in terms of stability, vulnerability, and balance of the network structure in the ancient city of Suzhou and proposed optimization measures [4]; Yu Sihan matched the data of urban functional areas and population spatial distribution by way of cellular signaling data to project the changes in the spatial and temporal distribution of the population; on this basis, the urban emergency shelters for earthquakes should be re-planned in accordance with the principles of accommodation, accessibility, and proximity to address the unreasonable layout of emergency shelters [5].

Two indices are often used in the evaluation of the spatial distribution rationality of urban parks, namely, the service area ratio and service population ratio. The accuracy of these two indices has been proven in previous studies [6]. Meanwhile, in order to increase the preciseness of the study as well as the service efficiency and economy of the spatial layout, another two indices are supplemented, namely, the service capacity ratio and service overlap rate. The four indices constitute the corresponding evaluation indices of the spatial distribution of urban emergency shelters.

At present, studies on the spatial layout of emergency shelters in China have mainly focused on the location selection of accessibility, and quite little has been done on sustainable disaster reduction, which is a disaster reduction model that is integrated into the overall environment and involves the whole process and multiple types of disasters. With the Xuzhou central urban area as the research object, this study used the ArcGIS10.8 network analysis module to optimize the spatial layout of emergency shelters in accordance with the four indices, namely, the service area ratio, service population ratio, service capacity ratio, and service overlap rate. This study provides new ideas and methods for research on the sustainable development of urban emergency shelters, which offers reference for the optimization of the spatial pattern of urban emergency shelters in Xuzhou and a scientific basis and suggestions for government decision-makers.

## 2. Research Area and Data Source

### 2.1. Overview of the Research Area

Xuzhou is listed as a national key city for flood prevention and earthquake relief [7]. Many disasters such as karst ground collapse, coal-mined ground collapse, and quarry landslides have occurred in Xuzhou, causing serious consequences including the deformation and subsidence of railroad roadbeds, and the subsidence and collapse of buildings [8]. The aforementioned geological hazards have become constraints to Xuzhou construction and the sustainable development of the national economy, not only affecting the sustainable development of the area, but also seriously threatening the lives and property safety of urban and rural residents. 

The research area is mainly the planning scope of the central urban area of Xuzhou, as determined by the revised “The Overall Urban Planning of Xuzhou (2007–2020)” in 2017, covering the administrative jurisdictions of Gulou District, Yunlong District, Quanshan District, the former Jiuli District, and the urban area of Tongshan District, with a total area of about 573.19 km^2^ (Figure 1). As shown in “The Bulletin of the Seventh National Census of Xuzhou”, the permanent residents within the research area totaled 1,798,800 at 00:00 on 1 November 2020. Currently, 22 emergency shelter and evacuation sites (parks, greenbelt, squares, etc.) have been built, with a total area of 5,200,000 square meters, accounting for 0.91% of the research area(Figure 2). The effective shelter area is about 250,000 square meters, and its capacity is about 110,000 people. Among them, nine are in Quanshan District, eight in Gulou District, four in Yunlong District, and one in the Economic Development District.

### 2.2. Data Source

The research data were divided into two main categories: geospatial data and demographic data. The geospatial data came from the Geographic Data Sharing Platform of the Chinese Academy of Sciences including administrative division, road networks and remote sensing images, etc. The POI data of parks, schools, and sports venues came from the open platforms of Gaode Map and Baidu Map, and were corrected by matching with the actual satellite maps. The demographic data were obtained from the Xuzhou Bureau of Statistics, “Xuzhou Statistical Yearbook 2020”, and “the Bulletin of the Seventh National Population Census of Xuzhou”.

## 3. Research Design and Model Construction

### 3.1. Research Design

By way of the GIS-network analysis, the spatial analysis module and network analysis module of GIS were mainly used to analyze the spatial layout of the emergency shelters.

Since urban emergency shelters have the usual characteristics of small scale, large number, and wide distribution, this study did not explore the specific spatial distribution, but only proposes planning guidance. The focus of this study is directed toward the spatial layout of the central and fixed emergency shelters (Table 1).

The research was designed in three steps:

Step 1: To determine the “service demand points” and “alternative facility points” for emergency shelters. “Service demand point” is the gathering point of citizens for their daily activities such as residence, employment, and life, and is also the starting point for travel during disaster prevention and avoidance; “alternative facility point” is the possible site for emergency shelters, and those sites that meet the basic conditions of disaster shelters can be used as alternative.

Step 2: To construct the location models of the emergency shelter, analyze and screen out the distance costs of the “service demand point” and “alternative facility point”, and propose the priority of the site selection of emergency shelters.

Step 3: To make planning suggestions to optimize the spatial layout based on the site selection priority and other conditions.

### 3.2. Model Construction

After World War II, with the emergence of the welfare state, Western governments involved themselves extensively and deeply in public economic activities by providing public facilities (services). It is against this background that the location theory of public facilities was developed. This theory is indeed based on location theory, which has experienced three stages: classical location theory, modern location theory, and modern location theory. The location theory of public facilities belongs to the research content of modern location theory [9].

By nature, the site selection of emergency shelters can be classified as the site selection of public facilities. Public facilities refer to all kinds of public service facilities that provide public services for residents. They have the characteristics of public (or quasi-public) property rights. The goal of public facilities is not profit, but mainly the optimization of social equity and interests. Therefore, non-profit and government investment in public facilities determine that the location theory of public facilities is different from that of traditional location theory. The location theory of public facilities and the traditional location theory were compared in the following four aspects: theoretical basis, location selection target, decision-maker, and research focus. As the results show, the theoretical model of public facility location theory is more service-oriented and more equitable and efficient. It is a government-led decision-making theory, and the research results are more accurate. Therefore, this study chose the location theory model of public facilities as the research basis.

#### 3.2.1. Construction of Location Model of Fixed Emergency Shelters

The emergency shelters that have been built should meet all of the necessary shelter requirements for victims and the site selection should adhere to the basic principles of fairness and efficiency. The number of facilities is to be reduced on the premise that all of the demand points have been covered; and a model for selecting shelter sites is to be constructed. The construction of this model consists of two stages. First, to build a collection of coverage. Second, to specify the minimum number of facilities to be used after fully covering all demand points, and then to use the median problem model to minimize the distance costs between the demand points and facility points by setting their distance. The following is the mathematical model:

Model assumptions: ① The location distribution of the candidate shelter facility points and emergency shelter demand points are known, and there is a certain distance between these points. ② The distance from the candidate shelter facility to the emergency shelter demand point is known, and the distance between the facility point and the demand point is based on the path distance of the actual road. ③ The demand point of the emergency shelter cannot be covered by two or more emergency shelter facility points, and there is only one facility point among the demand points in the covered area.

Let $H=\left\{H_{i}|i=1,2,\ldots ,m\right\}$ represent the collection of demand points and $K=\left\{K_{j}|j=1,2,\ldots ,n\right\}$ represent the collection of alternative facility points; a maximum range of r = 3000 m between the fixed emergency shelter and the demand point was set. $d_{ij}$ is the actual distance (road travel distance) between the demand point $H_{i}$ and the candidate shelter facility point $K_{j}$; ${Τ}_{i}=\left\{j|d_{ij}\leq r\right\}$ denotes the collection of candidate service facilities for demand point *i*. If the candidate service facility $K_{j}$ is determined as the service facility, $x_{j}=1$; otherwise, $x_{j}=0$.

The model Is constructed as:(1)$$Minz=\sum_{j\in K}x_{j}$$

Constraint conditions:(2)$$s.t.\sum_{j\in {Τ}_{i}}x_{j}\geq 1,\forall_{i}\in H$$
(3)$$x_{j}\in \left\{0,1\right\},\forall_{i}\in K$$
(4)$$0\leq d_{ij}\leq 3000$$

Equation (1) is the objective function, which is the minimum number of all facility points in the emergency shelter. Equation (2) is the constraint condition, based on which the radiation interval that meets the demand points of this emergency shelter is specified. Through the constraint condition, it can be guaranteed that the $x_{j}$ values are integers, and through the constraint condition calculated in Equation (4), it can be seen that the distance between the site facility point *j* and the emergency shelter demand point *i* was less than 3000 m. 

After modifying, the model can be expressed as:(5)$$Minz=\sum_{j\in K}d_{ij}x_{j}$$

#### 3.2.2. Construction of Location Model of Central Emergency Shelters

The central emergency shelter is the highest-level shelter, featured with large investment and construction costs. Its planning and construction are upgraded on the basis of fixed emergency shelters. The selection of its location should maximize the site service and focus on the use efficiency of the site. The model construction is divided into two stages. First, on the basis of the set covering model, the minimum number of facility points is selected when the demand points in the research area can be covered by all the facility points. Second, the E facility points are selected, and the maximization coverage model is used to obtain the solution; then, the facility points can cover the most demand points of the emergency shelter by setting the corresponding constraint conditions, which are shown in the mathematical model as follows: the mathematical model of the fixed emergency shelters and the corresponding model assumption conditions.

Let $H=\left\{H_{i}|i=1,2,\ldots ,m\right\}$ represent the collection of demand points and $K=\left\{K_{j}|j=1,2,\ldots ,n\right\}$ represent the collection of alternative facility points; $a_{i}$ is the demand of the demand point $H_{i}$ (such as population size, sales volumes and so on). Set a maximum range of r = 8000 m between the fixed emergency shelter and the demand point. $d_{ij}$ is the actual distance (road travel distance) between the demand point $H_{i}$ to the alternative shelter facility point $K_{j}$; E is the number of planned central emergency shelters (e≤n); ${Τ}_{i}=\left\{j|d_{ij}\leq r\right\}$ is the collection of alternative fixed emergency shelter facilities for demand point *i*. If demand point $H_{i}$ is covered by the emergency shelter, $y_{i}=1$; otherwise, $y_{i}=0$. If the candidate service facility $K_{j}$ is determined as an emergency shelter, $x_{j}=1$; otherwise, $x_{j}=0$.

The model is constructed as:(6)$$Maxz=\sum_{i\in D}a_{i}y_{i}$$

Constraint conditions:(7)$$s.t.\sum_{j\in {Τ}_{i}}x_{j}\geq y_{i},\forall_{i}\in H$$
(8)$$\sum_{j\in K_{}}x_{j}=e$$
(9)$$x_{j},y_{j}\in \left\{0,1\right\},\forall_{i}\in H,\forall_{i}\in K,$$
(10)$$0\leq d_{ij}\leq 8000$$

The objective function (Equation (6)) allows the facility points to cover the maximum emergency shelter demand points. Meanwhile, the demand of the demand points is also considered. The constraint condition (Equation (7)) is that when demand point *i* is assigned to cover, there is one and only one candidate shelter being used as the central emergency shelter among all candidates within the constrained distance. The constraint condition (Equation (8)) specifies that the number of selected central emergency shelters is E; the constraint (Equation (9)) is to ensure that $x_{j}$,$y_{j}$ is an integer; and the constraint condition (Equation (10)) is to determine that the distance from the emergency shelter demand point *i* to the site facility point j is within 8000 m.

The location models of the fixed and central emergency shelters were applied to the actual research, respectively. In fact, it was difficult to directly apply the models to the location selection of emergency shelters. Certain mathematical methods can be used to solve them, but the workload could be tremendous. Therefore, the facility location model can be solved through GIS technology and can be visually represented from the map. In the actual layout planning, the constraint conditions of different cities should also be considered, and the theoretical model obtained by the location model should be further adjusted and optimized.

## 4. Optimization of Spatial Layout of Emergency Shelters

### 4.1. Data Preparation and Analysis of Emergency Shelters

#### 4.1.1. The Analysis of Demand Points

GIS was applied in the research area to produce a corresponding regular grid, and based on the combination of urban road network density and the accuracy of ArcGIS calculation results, the city was divided into cell grids (300 m × 300 m). Then, the grid intersection points were selected as the center points of different areas, and the data of the cell intersection points in roads, non-construction land, and water were removed after screening. Combined with the actual planning layout of the site and with reasonable adjustment, all center points in the construction site that met the conditions were selected, with a total of 4285 (as shown in Figure 3).

#### 4.1.2. The Analysis of Candidate Facility Points

For the selection of emergency shelters, existing resources such as parks, greenbelts, squares, stadiums, and school playgrounds can be utilized as candidate sites. By collecting data related to various emergency shelters in Xuzhou including existing park, greenbelts, other green squares, sports sites, and school playgrounds in the city that can be used as emergency shelters for usual disaster prevention, it was found that there was a total of 367 candidate sites, 294 of which (Figure 4) were finally screened out after analyzing their safety, floor space, and traffic.

### 4.2. Optimization of Spatial Layout of Fixed Emergency Shelters

The selection of fixed emergency shelter sites is based on the site selection model that has been constructed. In other words, on the basis of the minimum number of facilities determined under the condition of the full coverage of demand points, the median problem model was used to minimize the total cost between the facility points and the demand points by setting distance restrictions [10]. First, a new location assignment model was built on the basis of the screened candidate emergency shelters, then the 294 screened candidate shelters were loaded as facility points and all demand points as request points. In the model operation, the actual length of roads in the central urban area of Xuzhou was taken as the travel cost, the impedance interruption was set to 3000, that is, the maximum radiation radius of fixed shelters was 3000 m; and the impedance transformation was linear by default; based on the above steps, the solution was conducted. First, the minimization facility point model was applied to the candidate shelters. After the solution and calculation, the number of fixed emergency shelters for earthquakes required was at least 79, and the number of demand points covered was 4055, with a coverage rate of 94.63%. The operation results of the model are shown in Figure 5.

If the principles of reasonableness and fairness of seismic evacuation are taken into account, it is necessary to cover all demand points in the area, and at the same time, all of the demand points are placed in the emergency seismic shelter and evacuation area at the travel cost required by the index. However, the actual situation shows that, in general, the population density in the central area of the city is higher, while it is lower in the peripheral areas of the city. The population distribution is relatively scattered, and in general, the distribution of urban population is unbalanced. Therefore, if the evacuation places are evenly distributed throughout the area only considering the distance cost, it will lead to low utilization of evacuation places in marginal areas. Economically, this layout does not make sense.

Therefore, in order to achieve the fairness and economy of fixed evacuation sites, the minimum impedance model was again operated to confirm the number and location of fixed evacuation sites based on the minimization facility point model. The number of fixed evacuation sites before each operation of the minimum impedance model was given and the actual road distance was used as the operation cost. The number of facilities was reduced from 79 to 73 by minimizing the number of facilities and the arithmetic results of the number of fixed evacuation sites, that is, the coverage rate and shelter area, were obtained. The green dot blocks in the figure show the decreasing candidate points (Figure 6 and Figure 8). Table 2 summarizes the relationship between the number of fixed evacuation sites and their coverage rate.

The variation in the coverage rate with the number of sites obtained from the above analysis was plotted as a scatter plot and simulated as a curve. The results are shown in Figure 7.

As can be seen from the figure, when the number of places was 73 (Figure 8), the curve took a turn and the curvature had a significant decrease with a coverage rate of 94.31%. After that, the slope of the curve tended to be smooth again. Taking the rules of fairness and economy into account and combining the coverage rate of evacuation places, the evacuation area, and other hazard reasons, the number of fixed evacuation places was set at 73.

### 4.3. Optimization of Spatial Layout of Central Emergency Shelters

The central emergency shelter is the highest-level shelter with the highest planning and construction cost. Accordingly, the main considerations in its planning and construction are economy and efficiency. This was selected on the basis of the optimized fixed emergency shelters. The constructed location model of the central earthquake emergency shelter was selected for the solution, that is, the minimization facility point model was run first, and then the maximum coverage model was run [11]. First, the minimized facility point model in the network analysis location assignment model in ArcGIS 10.5 was used to process the data and the demand points in the research area that could be covered by the facility points of the central shelters were selected. In the model operation, the actual length of the roads in the central urban area of Xuzhou was used as the travel cost and 27 candidate points that met the requirements of central shelters were sorted as the facility points. Then, the minimization facility point model was run with the impedance interruption of 8000 m. The operation results of the model are shown in Figure 9 The minimum number of facility points was 12, and the number of demand points covered was 4131, with a coverage rate of 96.41%.

The central emergency shelter is often used as the command center of urban earthquake relief, which not only possesses the basic functions of fixed emergency shelters, but also serves as the place for receiving and distributing rescue supplies and the command place for health and epidemic prevention. In its planning and construction, the construction cost should be minimized while considering the efficiency [12]. On the basis of the 12 candidate central emergency shelters for earthquakes selected above, different numbers of facilities were set as E (E is a positive integer less than 12), and the maximization coverage model was run to solve them, respectively, as shown in Figure 10 and Figure 11.

As the solution showed (Table 3), when E = 10, the number of demand points covered was 4119 and the coverage rate was 96.13%; when E = 9, these were 4097 and 95.61%; when E = 8, these were 4046 and 94.42%; when E = 7, these were 3919 and 91.46%. Therefore, eight facility points were selected as the central emergency shelters, which ensures the economy and fairness of building shelters.

## 5. Rationality Evaluation of Spatial Layout of Emergency Shelters

### 5.1. Rationality Evaluation Indexes of Spatial Layout of Emergency Shelters

In this paper, four evaluation indices, namely, the service area ratio, service population ratio, service capacity ratio, and service overlap rate were used to evaluate the rationality of the spatial layout of emergency shelters in the central urban area of Xuzhou.

The service area ratio is the ratio of the total service area of the public service facility to the total area of the research area. It is a reflection of the space service capacity of public facilities.

The service population ratio is the ratio of the number of people in the service area of a public service facility to the total population in the research area. This is a reflection of the population capacity and service capacity of public facilities.

The service capacity ratio is the ratio of the total population in need of public service facilities in the research area to the total design population capacity of public service facilities. This is a reflection of the space service efficiency of public service facilities.

The service overlap ratio is the ratio of the area of the overlapping part of the coverage of each public service facility and other public service facilities to the total area of the coverage of all shelters. This is a reflection of the overlapping spatial layout of public service facilities. 

### 5.2. Rationality Evaluation of Spatial Layout of Emergency Shelters

According to the built road network, the accessibility of emergency shelters in Xuzhou was first evaluated after the optimized layout. The maximum service distance of fixed emergency shelters was 3000 m, and that of central emergency shelters was 8000 m. Seventy-three selected fixed emergency shelters were analyzed, and the service area function of the ArcGIS network analysis can be used to calculate the service area of the shelter. Based on the road network model, the coverage area of facilities within the service radius can be more accurately simulated on a realistic road network based on the traffic distance. For fixed and central shelters, the impedance was set to the road network travel cost in the analysis setting of the new service area layer attribute, and the default interruption was set to 3000 and 8000, respectively. The solution results are shown in Figure 12 and Figure 13. The same method was applied to solve the completed emergency shelters and the results are graphically represented (Figure 14).

The above analysis of the reasonableness of the layout of the candidate central emergency shelters in Xuzhou was carried out by arithmetic and calculation of the service parameters of the evaluation criteria statistics (Figure 15). The results are shown in Table 4 and Table 5.

#### 5.2.1. Analysis of Service Index Parameters

The analysis was based on the statistical comparison of the service indices of the built emergency shelters and the emergency shelters after layout optimization. The overall comparison showed that the service area ratio of emergency shelters before and after optimization was 27.28% and 67.96%m respectively, and the service population ratio was 43.93% and 80.96%, respectively. It can be seen that the number of emergency shelters before optimization was too small and far from meeting the demand for shelters in the central urban area of Xuzhou, and the service area and population of emergency shelters after optimization have been greatly improved. The comparison of the service area ratio and the service population ratio of emergency shelters before and after optimization are represented by bar graph plots as shown in Figure 16 and Figure 17, respectively.

From the comparison of the service overlap rate, the service overlap rate of emergency shelters before optimization was high, while their service area and population ratio were extremely low. The number of optimized emergency shelters increased and the service overlap rate also increased. However, the service area and population were substantially improved, with a 43.09% increase in the former and 38.05% in the latter. The reason for the high service overlap rate is that the built emergency shelters are too concentrated, mainly in Quanshan District and Gulou District, and they are close to each other; another reason is that the population in the central urban area is large and concentrated and enough emergency shelters must be arranged to meet the needs of the residents. Meanwhile, for the random distribution and fixed places of the emergency shelters, the phenomenon of service overlap will inevitably arise when screening is carried out. In order to balance equity and efficiency, it shows that the optimized emergency shelter is more reasonable.

#### 5.2.2. Macro Analysis of the Central Urban Area of Xuzhou

Analysis of Table 5 shows that the service shelter population of emergency shelters in the central urban area of Xuzhou reached 80.96%. In general, these shelters covered a wide population with a reasonable layout and can meet the index of covering more than 80% of the population. In terms of the service area ratio, the ratio in the center was 67.96%, which can satisfy about 70% of the demands in the research area. In terms of the service area ratio of each district, the ratios of Quanshan District, Gulou District, and Yunlong District were all above 70%. In contrast, those of Tongshan District and the Economic Development District were relatively low, between 50% and 60%, which correspond to the actual situation of low population density and low construction density of Tongshan District and the Economic Development District. In general, it shows that the planning layout of the emergency shelter meets the demand. In terms of the service capacity ratio, this was generally 130.62%, which indicates that there is a surplus in the service capacity of shelters and can meet the demand of urban residents for shelters. In terms of the service overlap rate, the average value was 86.67%, while the coverage rate reached 94.31%, where it can be seen that the optimized spatial layout of the emergency shelters was more reasonable.

#### 5.2.3. Analysis of Each District

For each district in Xuzhou, the service area ratios in Quanshan District, Gulou District, and Yunlong District in the central urban area were relatively high due to the large number and the dense distribution of emergency shelters. 

In terms of the service population ratio, those for the Economic Development District and Tongshan District were relatively low compared with other districts, both between 50% and 60% because of their relatively small population size, small population density, low development intensity, and a large number of rural populations. The population in these areas is sparse and relatively scattered, and the area beyond the service radius fails to enjoy the corresponding services. Furthermore, there are still some deficiencies in the distribution of emergency shelters after layout optimization. 

In terms of the service capacity ratio of each district, those of all the other districts, except for Gulou District, exceeded 1, which means that the design capacity of the other four districts is enough to meet the evacuation and shelter requirements of the population in the area. However, the design capacity of Gulou District cannot fully accommodate the shelter population, because the effective shelter area of the emergency shelters built in Gulou District is small and needs to be further improved.

### 5.3. The Supporting Strategy for Emergency Shelters

Site selection should be in accordance with the relevant government laws and regulations as well as local standards. At present, China has promulgated a number of laws and regulations related to urban disaster prevention and reduction. Local governments at all levels should constantly supplement and improve the planning, construction, and design standards of earthquake emergency shelters in accordance with the rules and regulations promulgated by the state and the conditions of their own regions. Meanwhile, the site selection of emergency shelters should closely match the policies and relevant standards formulated by the government.

### 5.4. Innovation Points

This paper constructed a location selection model system of urban emergency shelters that can apply the Network Analyst extension module in ArcGIS (minimize resistance and maximize coverage). The optimization of the layout of emergency shelters in downtown Xuzhou was completed. The rationality evaluation index was constructed to verify the scientificity and effectiveness of the location model, providing guidance for the planning and construction of shelters in the future.

## 6. Conclusions

There are many service blind areas and service overlap areas in the emergency shelter service areas. Emergency shelters are basically places with other urban functions that are “borrowed temporarily”. In this sense, their layout follows their own principles. For example, city parks have a different service radius according to their level; primary and secondary schools are distributed according to the service requirements of residential areas; and city squares, parking lots, sports fields are allocated according to the layout of the city’s functional needs. The layout of these sites does not need to consider the layout of other sites. For instance, schools may happen to be located next to a park or square. However, if they are also used as emergency shelters, the service area would overlap. Therefore, the government should determine the service areas of emergency shelters according to the accessibility evaluation of people, and supplement more emergency shelters in the service blind areas. Meanwhile it should find out and reduce the service overlapping areas by re-siting in order to avoid waste.The layout of the emergency shelter failed to match the demand for shelter. The analysis of the layout of emergency shelters points out some existing problems such as the confusion in the scale level of shelters, uneven construction in each district, and insufficient number of places built in the city. In China, the selection of urban emergency shelters generally follows the principle of dual use for daily life and disaster relief. With urban parks, squares, parking lots, sports fields, school playgrounds, and other sites as shelters, although these sites are combined to some extent with the distribution of residential areas in their layout, their service population is often calculated on the basis of the usual population. In the event of a disaster, each area has different disaster risks. Although old urban communities, urban villages, etc. have about the same population as new residential areas, these communities are often more severely damaged after a disaster and have a greater need for disaster avoidance. Local governments should improve urban emergency shelter systems and focus on disaster damage assessment in cities to improve the sustainability of urban disaster reduction.Regular evacuation drills should be organized for residents to prevent disasters. With the knowledge of the public, all construction work can play a role. In many cities in China, residents know very little about how to respond to disasters, which requires the government and communities to develop perfect emergency plans. All residents should follow the procedures for self-rescue, and arrive at the emergency shelter along the evacuation route. Through long-term training, residents should be familiar with the evacuation plans and evacuation routes for different disasters around their homes, so that they can save themselves and each other in an orderly manner during disasters, minimize casualties, and play the life-saving role of emergency shelters and emergency evacuation routes.

Although this paper has made some achievements in the location optimization and layout of public facilities in urban emergency shelters, there are still some limitations which are hoped to be further improved in the future. (1) This study took the central urban area of Xuzhou as the research object to explore the layout optimization of urban emergency shelters. Xuzhou, as an inland city with little relief, is featured with flat terrain. In this sense, this study is of certain instruction to cities with flat terrain. However, the special cases of mountain cities, coastal cities, and cities across rivers were not considered in this study. We expect to study urban emergency shelters for different types of cities in the future. (2) Currently, the site selection of emergency shelters is mainly focused on parks, schools, sports venues, and other places and deals with common public emergencies and natural disasters such as earthquakes, fires, floods, and epidemics. However, it is far from competent to deal with other man-induced disasters such as war, toxic chemical leakage, nuclear leakage, and other disasters. Future research should cover more types of disasters and enrich the types of emergency shelter site selection to greater meet the shelter needs of residents.

## Figures and Tables

**Figure 1 ijerph-20-02127-f001:**
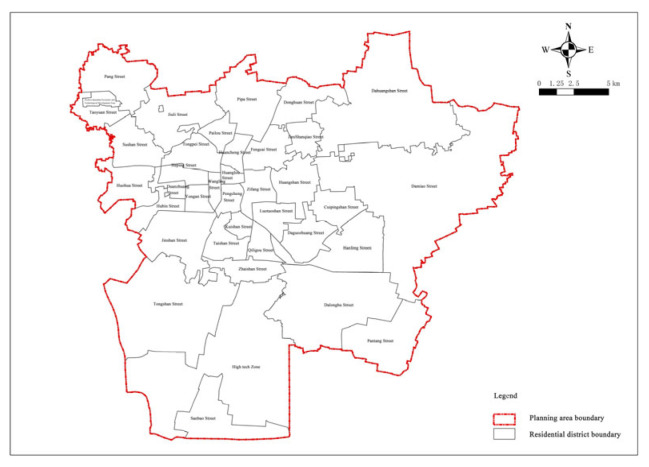
Planning scope.

**Figure 2 ijerph-20-02127-f002:**
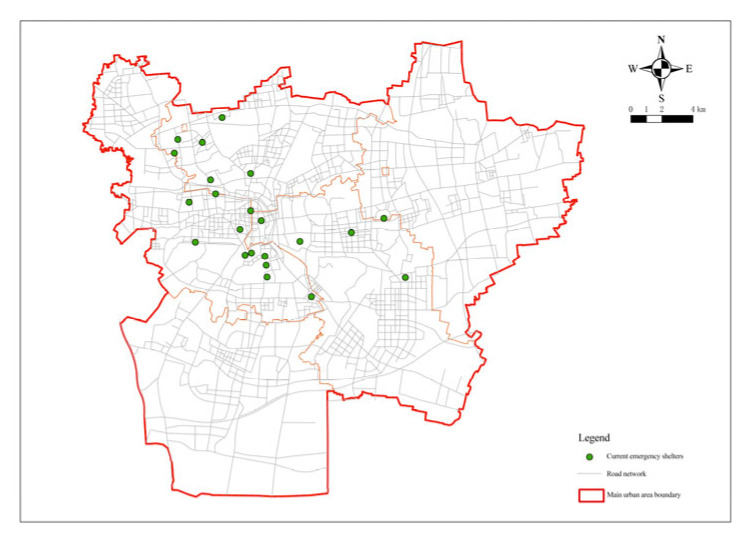
Current emergency shelters.

**Figure 3 ijerph-20-02127-f003:**
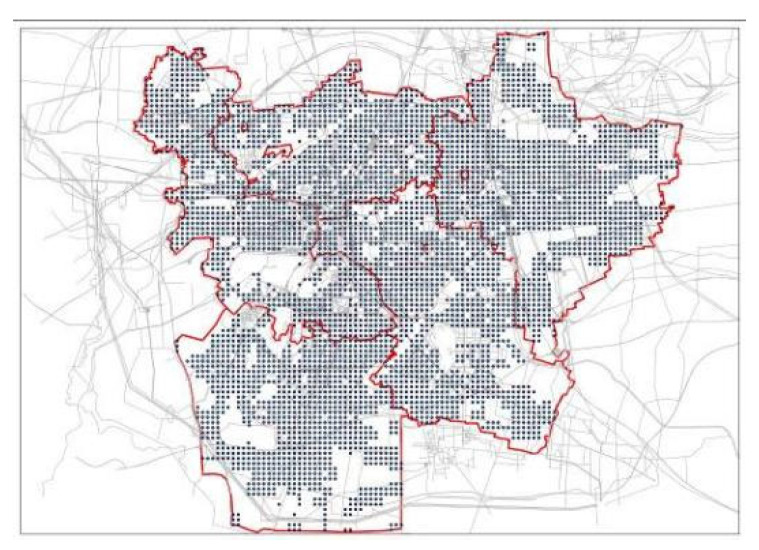
Distribution of the demand points.

**Figure 4 ijerph-20-02127-f004:**
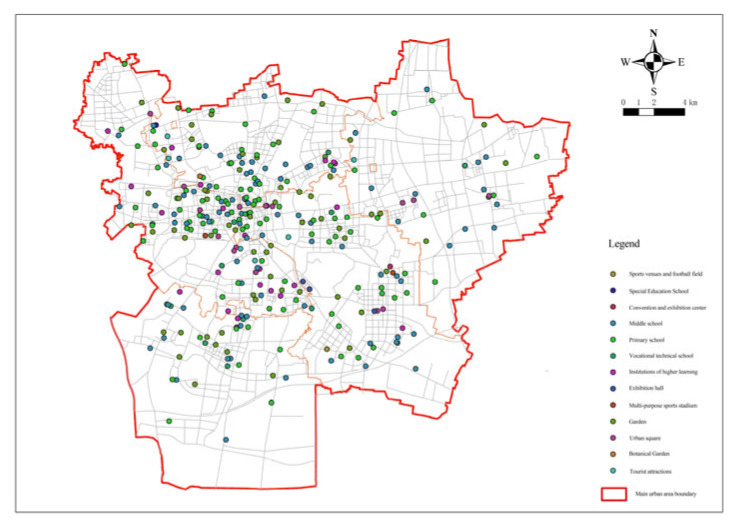
Distribution of the candidate facility points.

**Figure 5 ijerph-20-02127-f005:**
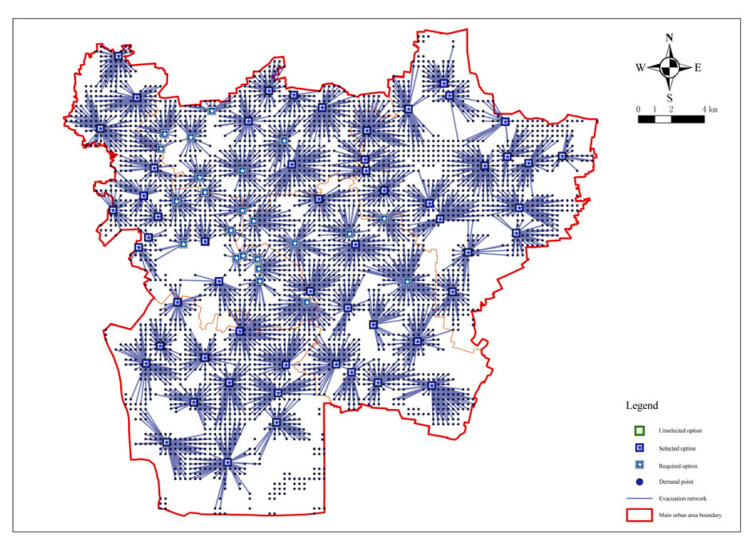
Analysis of the minimum facilities of fixed emergency shelters.

**Figure 6 ijerph-20-02127-f006:**
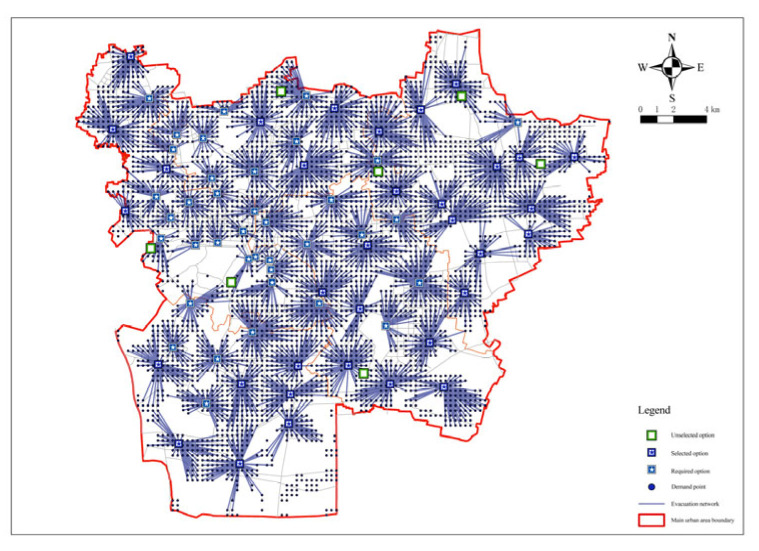
Minimization impedance analysis for a facility point resistance of 72.

**Figure 7 ijerph-20-02127-f007:**
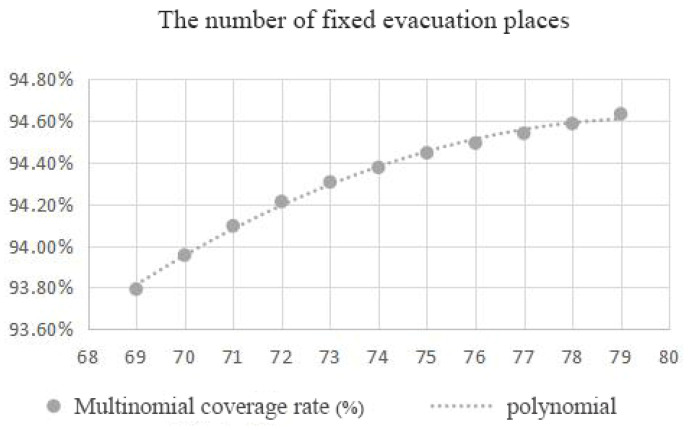
Relationship between number of fixed sites and coverage.

**Figure 8 ijerph-20-02127-f008:**
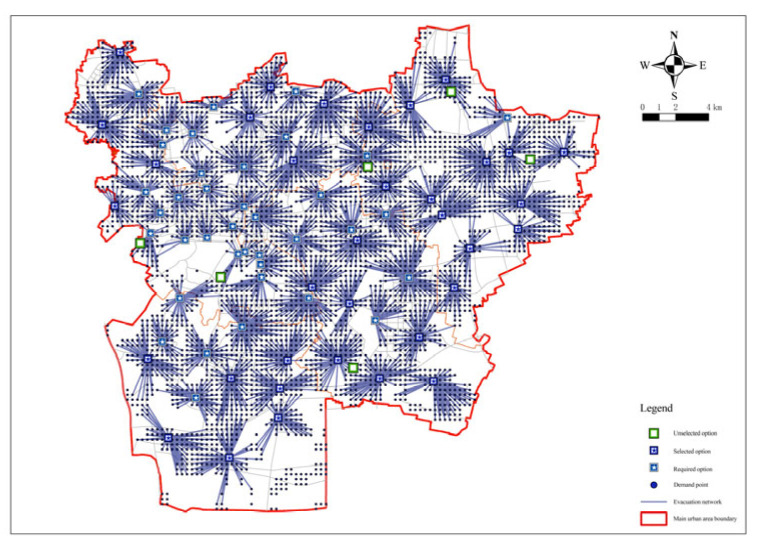
Minimization impedance analysis for a facility point resistance of 73.

**Figure 9 ijerph-20-02127-f009:**
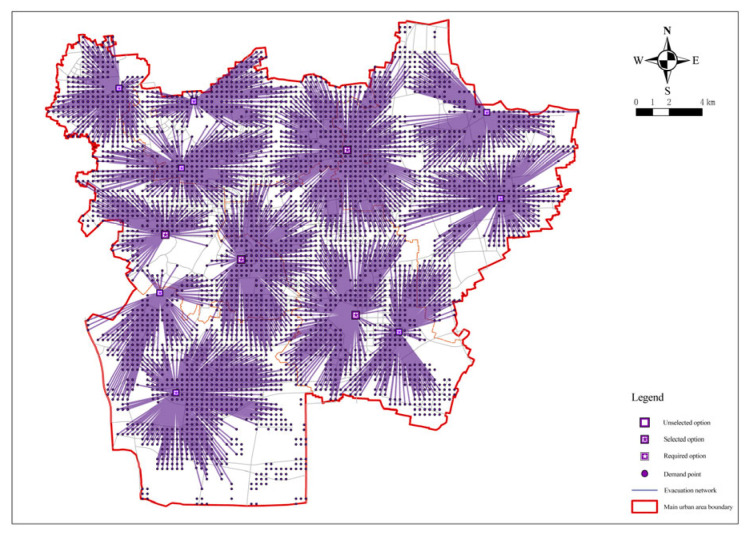
Minimum facilities analysis of the central emergency shelters.

**Figure 10 ijerph-20-02127-f010:**
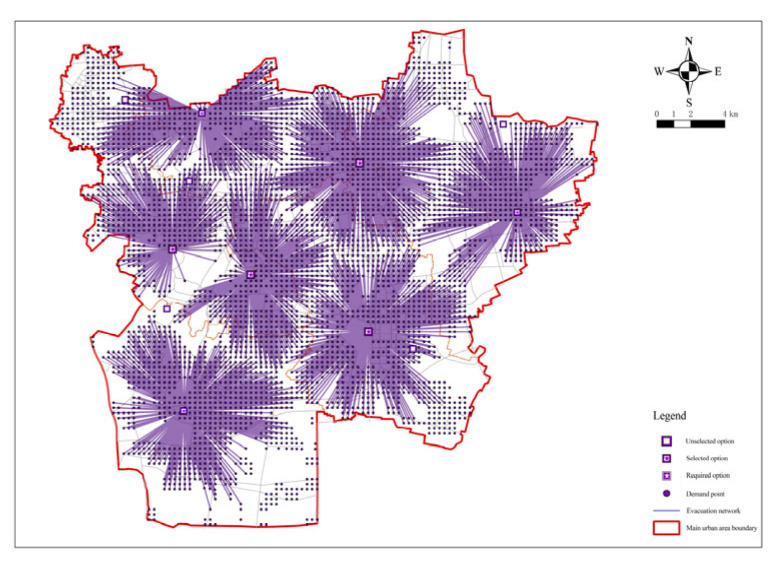
Analysis of the maximum coverage of E = 7.

**Figure 11 ijerph-20-02127-f011:**
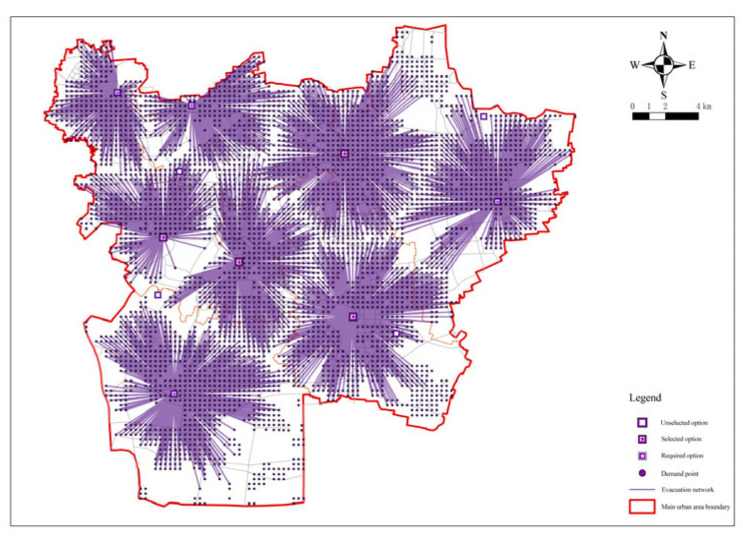
Analysis of the maximum coverage of E = 8.

**Figure 12 ijerph-20-02127-f012:**
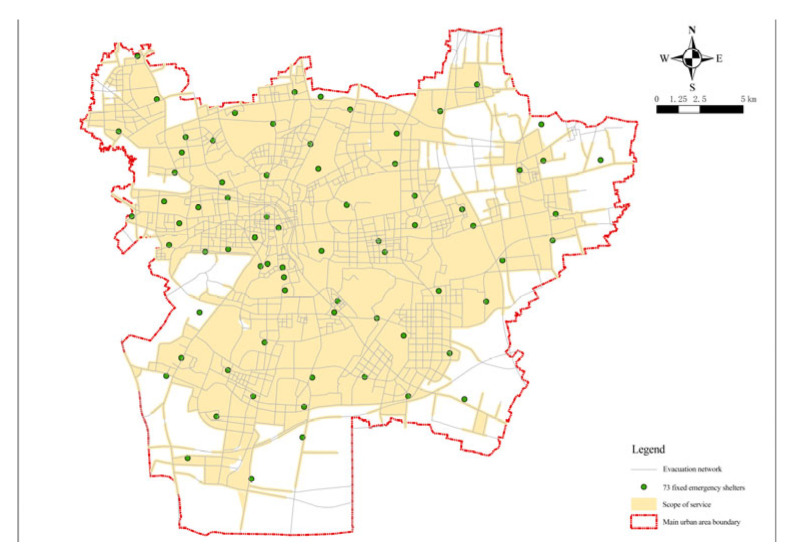
Fixed emergency shelters with the service area of 3000 m.

**Figure 13 ijerph-20-02127-f013:**
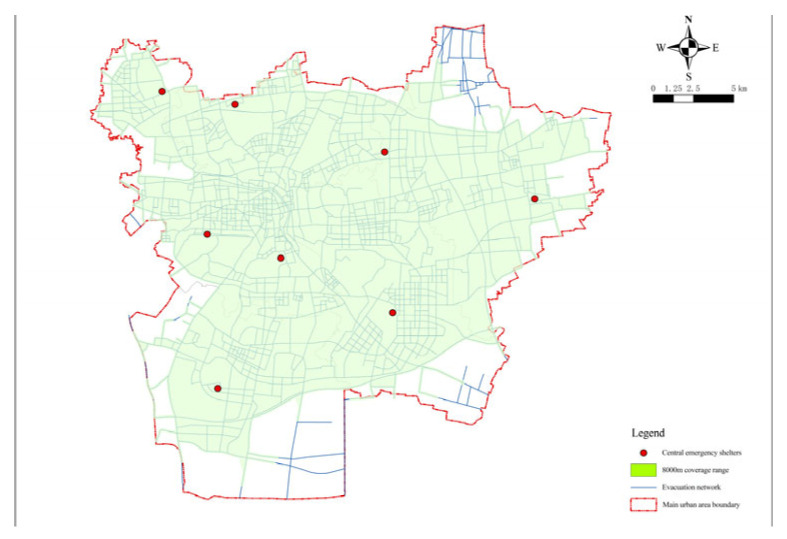
Central emergency shelters with the service area of 8000 m.

**Figure 14 ijerph-20-02127-f014:**
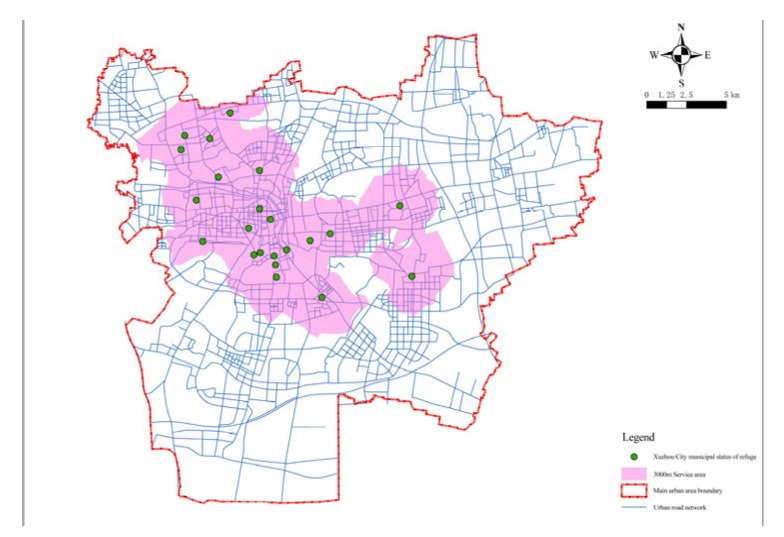
Built emergency shelters with the service area of 3000 m.

**Figure 15 ijerph-20-02127-f015:**
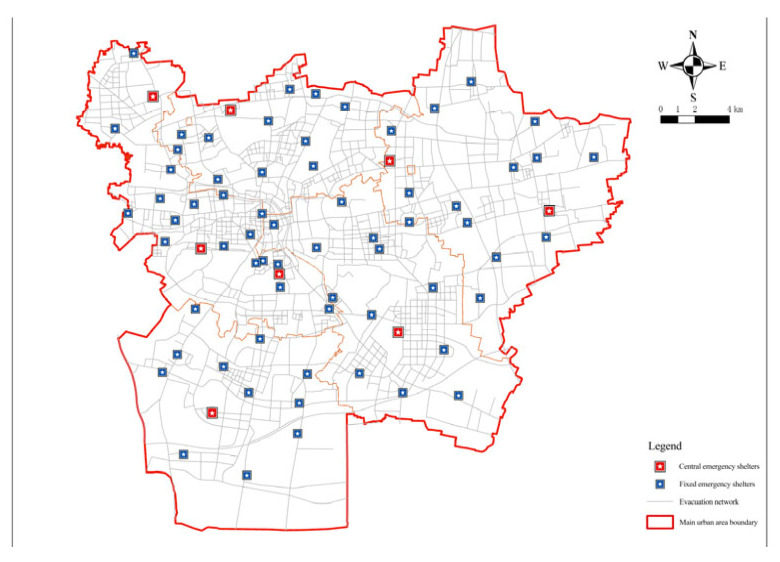
Optimized layout of emergency shelters.

**Figure 16 ijerph-20-02127-f016:**
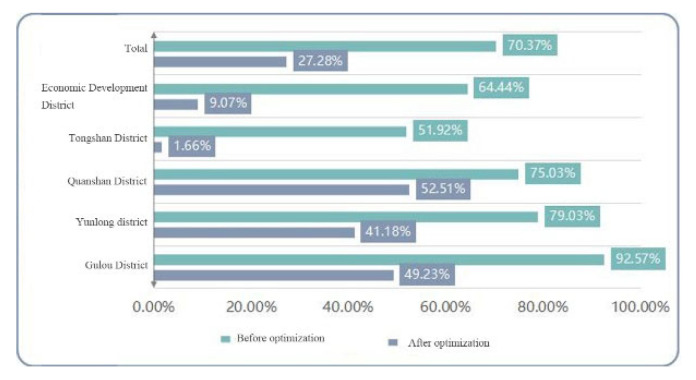
Bar chart of the service area ratio before and after optimization.

**Figure 17 ijerph-20-02127-f017:**
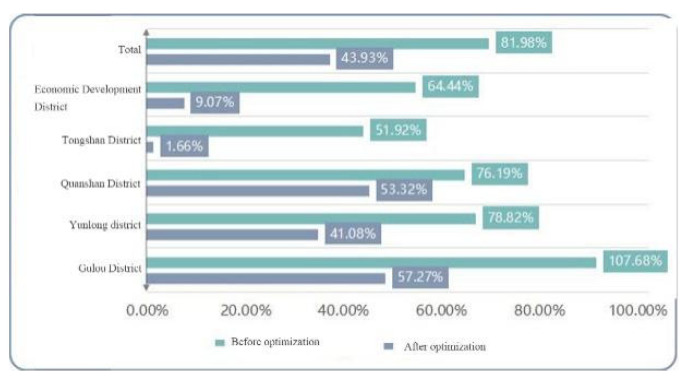
Bar chart of the service population ratio before and after optimization.

**Table 1 ijerph-20-02127-t001:** Planning and construction requirements of various urban emergency shelters.

Project	Nature of Shelters	Floor Space (hm^2^)	Per Capita Effective Shelter Area (m^2^)	Service Radius (km)	Walking Time (min)	Effective Width of Evacuation Channels	Essential Facilities
Emergency shelters	Temporary	≥0.1	≥1	0.5	≤10	≥4	---
Fixed emergency shelters	Middle or short term	1–20	≥2	2–3	≤60	≥7	Basic facilities
Fixed emergency shelters	Middle or long term	≥20 generally,	≥2	Referable fixed emergency shelters	Referable fixed emergency shelters	≥15	General facilities
		over 50					Integrated facilities

**Table 2 ijerph-20-02127-t002:** The number and coverage rate of fixed evacuation facilities in Xuzhou.

Number of Facilities (Unit)	Number of Demand Point in the Covering Point (Unit)	Total Number	Coverage Rate
69	4019	4285	93.79%
70	4026	4285	93.96%
71	4032	4285	94.10%
72	4037	4285	94.21%
73	4041	4285	94.31%
74	4044	4285	94.38%
75	4047	4285	94.45%
76	4049	4285	94.49%
77	4051	4285	94.54%
78	4053	4285	94.59%
79	4055	4285	94.63%

**Table 3 ijerph-20-02127-t003:** The number and coverage rate of central evacuation places in Xuzhou.

Number of Facilities (Unit)	Number of Demand Point in the Covering Point (Unit)	Total Number	Coverage Rate
4	2584	4285	60.30%
5	3143	4285	73.35%
6	3556	4285	82.99%
7	3919	4285	91.46%
8	4046	4285	94.42%
9	4097	4285	95.61%
10	4119	4285	96.13%
11	4129	4285	96.36%
12	4131	4285	96.41%

**Table 4 ijerph-20-02127-t004:** Statistical table of service indices of established emergency shelters in Xuzhou.

District Name	Service Area (ha)	Total Area(ha)	Service Area ratio	Service Population (Person)	Permanent Residents in the Area (Person)	Service Population Ratio	Service Overlap Rate	Service Capacity Ratio
Gulou District	38.10	77.40	49.23%	243,483	425,143	57.27%	76.59%	9.40%
Yunlong District	49.45	120.07	41.18%	193,704	471,566	41.08%	45.98%	5.01%
Quanshan District	53.52	101.92	52.51%	330,488	619,784	53.32%	77.37%	6.45%
Tongshan District	2.13	128.49	1.66%	1410	84,932	1.66%	4.74%	0.00%
Economic Development District	13.18	145.31	9.07%	17,081	188,320	9.07%	1.69%	5.63%
Total	156.39	573.19	27.28%	786,166	1,789,745	43.93%	53.27%	6.38%

**Table 5 ijerph-20-02127-t005:** Statistical table of service indices of optimized emergency shelters in Xuzhou.

District Name	Service Area (ha)	Total Area (ha)	Service Area Ratio	Service Population (Person)	Permanent Residents in the Area (Person)	Service Population Ratio	Service Overlap Rate	Service Capacity Ratio
Gulou District	71.64	77.40	92.57%	457,811	425,143	107.68%	96.83%	40.05%
Yunlong District	94.76	120.07	78.92%	371,170	471,566	78.71%	89.52%	123.28%
Quanshan District	76.48	101.92	75.03%	472,234	619,784	76.19%	91.81%	149.28%
Tongshan District	66.67	128.49	51.89%	44,071	84,932	51.89%	76.12%	298.75%
Economic Development District	79.99	145.31	55.05%	103,671	188,320	55.05%	82.24%	216.22%
Total	389.54	573.19	67.96%	1,448,957	1,789,745	80.96%	86.67%	130.62%

## Data Availability

The data used to support the findings of this study are available from the corresponding author upon request.

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
