# Peer review of "Spatial Layout Planning of Urban Emergency Shelter Based on Sustainable Disaster Reduction"

_ijerph, 2023, doi:10.3390/ijerph20032127_

Round 1
Reviewer 1 Report
"Spatial Layout Planning" is a fascinating manuscript and I recommend publication. Its strength is in drawing up ArcGIS and mathematical models to plot emergency shelter locations. The manuscript builds nicely upon previous research and the authors well support their conclusions. There are a few minor suuggestions. In your conclusion discuss why this is a superior model compared to what is being used at present to make site decisions. Two, how does this model build upon the current literature and what do we learn here that we did not know before? How useful is this model for different types of disasters and shelters and is the model applicable to other cities or types of political regimes. Finally, this model does not discuss the political forces that affect location decisions. Is that realistic or is the purpose to move beyond political considerations?
Reviewer 2 Report
This is a well designed research but two changes are needed.
1. on page 2 under Introduction in the second paragraph 2nd line- this should say "set of four indicators". Also these indicators are not explained or defined until page 10 under section 5.1. I think these definitions need to be moved to the introduction plus why they were chosen as the indictors used.
2. Under the conclusion section - the first paragraph is taken right from the paper template and need to be eliminated or reworked. Also the conclusion section is missing; the limitations of the research conducted, the generalizability of the research to other urban locations, and what future research is needed.
Reviewer 3 Report
The methodology and analysis of the study are properly developed, but the introduction is too rough. A more precise definition of the case study would be necessary in the title. After that, both in the abstract and in the introduction, the question and the expected result of the study should be stated more precisely. This is now only displayed very roughly. All of this should be formulated in connection with the conclusion. The bibliography and contextualization are very limited, which also highlights the problem of generalizing results based on the case study. The limitations of the results should be more clearly articulated.
